# Advances in Extraction, Purification, and Analysis Techniques of the Main Components of Kudzu Root: A Comprehensive Review

**DOI:** 10.3390/molecules28186577

**Published:** 2023-09-12

**Authors:** Tong Xuan, Yuhan Liu, Rui Liu, Sheng Liu, Jiaqi Han, Xinyu Bai, Jie Wu, Ronghua Fan

**Affiliations:** Department of Sanitary Inspection, School of Public Health, Shenyang Medical College, Shenyang 110034, China; xthhxx2022@163.com (T.X.); yu_hanliu@126.com (Y.L.); 13179007559@163.com (R.L.); 15841050435@139.com (S.L.); h15802463430@163.com (J.H.); 18766576032@163.com (X.B.)

**Keywords:** puerarin isoflavone, extraction, purification, analysis techniques

## Abstract

Kudzu root (Pueraria lobate (Willd.) Ohwi, KR) is an edible plant with rich nutritional and medicinal values. Over the past few decades, an ample variety of biological effects of Pueraria isoflavone have been evaluated. Evidence has shown that Pueraria isoflavone can play an active role in antioxidant, anti-inflammatory, anti-cancer, neuroprotection, and cardiovascular protection. Over 50 isoflavones in kudzu root have been identified, including puerarin, daidzein, daidzin, 3′-hydroxy puerarin, and genistein, each with unambiguous structures. However, the application of these isoflavones in the development of functional food and health food still depends on the extraction, purification and identification technology of Pueraria isoflavone. In recent years, many green and novel extraction, purification, and identification techniques have been developed for the preparation of Pueraria isoflavone. This review provides an updated overview of these techniques, specifically for isoflavones in KR since 2018, and also discusses and compares the advantages and disadvantages of these techniques in depth. The intention is to provide a research basis for the green and efficient extraction, purification, and identification of Pueraria isoflavone and offers investigators a valuable reference for future studies on the KR.

## 1. Introduction

Kudzu root (Pueraria lobate (Willd.) Ohwi, KR), a legume taxon indigenous to Southeast Asia, boasts a rich variety of species and can be found across China [1], including in Hebei, Guangxi, Anhui, Liaoning, and Shandong. In fact, it has become one of the most prevalent twining shrub plants in urban forests. In 2022, the Ministry of Health of China officially declared KR as a “medicine and food homology” plant, thereby expanding its use in health food.

Kudzu root is a traditional Chinese food and medicinal herb. It was commonly used in fresh food, stir-fried vegetables, soup, and tea. Pueraria-series products have developed in recent years due to their abundance of nutrients like starch, isoflavones, and dietary fiber, both domestically and internationally. At present, there are many explorations on the eating methods of KR. In addition to being used in meal replacement powders for brewing and eating, KR noodles, biscuits, cakes, steamed bread, and other flour products can also be made by combining KR with other botanical raw materials [2,3]. KR can be used to create a variety of tasty beverages [4,5,6]. Most importantly, as an excellent natural material for the development of new health foods, KR is extensively used in the food and health product industries and has great market potential. By searching the website of the China State Administration for Market Regulation using the keyword “Kudzu root”, 613 health food products containing KR with clear approval numbers were collected. There were 386 examples of KR health food in the form of capsules. KR health food was produced into 27 different types of tea, 9 different types of powder, and 6 different types of beverages.

The common ingredients of KR include flavonoids, starch, cellulose fiber, protein, etc. Among them, the best-effect ingredients are isoflavones. Isoflavones are part of a large family of secondary plant metabolites called flavonoids. They are dietary phytoestrogens occurring naturally in legumes. A reliable source of isoflavones is Pueraria lobata, according to pertinent research studies. The mass fraction of puerarin can range from 1.58% to 7.68%, whereas the overall mass fraction of isoflavones in Pueraria lobata can range from 6.2% to 17.0% [7,8]. Currently, it has been revealed that more than 50 different types of puerarin isoflavone have unambiguous structures, such as Puerarin [9], Daidzein, Daidzin, Puerarinxyloside [10], 3′-Hydroxy Puerarin, Genistein, Genistin, Biochanin A, Formononetin [11], 6″-O-Malonylgenistin [12], 6″-O-α-d-Glucopyranosylpuerarin [13], Calycosin [14], 4′-Hydroxy-7-Hydroxymethyl-6-Methoxyisoflavone [15], 4′,6-Dimethoxy-8-Hydroxy-7-Hydroxymethyl Isoflavone [16], 3′-MethoxypuerarinA, 3′-MethoxypuerarinB [17], etc.

The main isoflavones in KR are displayed in Table 1. The biological activity and mechanism studies of PR isoflavone are shown in Figure 1. They have demonstrated positive effects, including hepatoprotective [18,19], anti-inflammatory [20], anti-cancer [21,22,23], anti-osteoporosis [24,25,26], and protection for the heart and brain [27,28]. Puerarin can be used to reduce cardiac hypertrophy and arrhythmia [27,28,29,30] and prevent atherosclerosis [31,32]. Research has revealed that puerarin isoflavone has positive effects on common health issues like diabetes, hypertension, and hyperlipidemia. It has been established that puerarin, genistein, daidzein and daidzin are efficient treatments for hypertension [33,34,35,36,37,38]. Pueraria extract, according to Li et al., can lower blood pressure in mice fed a high-salt diet by modulating gut flora [37]. According to Yang et al., pueraria is effective in treating physical illnesses that co-occur with depression [38]. It covers post-stroke depression as well as depression caused by coexisting conditions, such as diabetes, coronary heart disease, migraines, Parkinson’s disease, and others. KR has the potential for an alcohol intake reduction [39] and, thus, in the long-term, prevention of alcohol addiction [40]. Zhou et al. gave alcoholic male Wistar rats puerarin, daidzein, and puerarin extract [41]. The findings demonstrated that daidzin and puerarin were able to dramatically raise the levels of two types of ghrelin, which in turn decreased alcohol consumption.

Effective and adequate procedures are required for the extraction, purification, and analysis of puerarin isoflavone. In recent years, a variety of methods for extracting flavonoids have been developed, such as organic solvent extraction, ultrasonic-assisted extraction, microwave extraction, and supercritical fluid extraction. Advances in science and technology have made it possible to identify puerarin isoflavone using more sophisticated instruments and multi-instrument combination detection methods, such as high-resolution mass spectrometry (HRMS), LC-MS, GC-MS, etc. This review provides an updated overview of these techniques for isoflavones in KR. A general outline of a sample flow chart covering the preparation, extraction, purification, and analysis process is represented in Figure 2. 

## 2. Sample Pretreatment Methods

Sample pretreatment refers to the separation of components from complex systems by enzymatic hydrolysis, extraction, enrichment, purification, and concentration. It is a process of removing impurity interference, increasing the concentration of the component to be tested, and facilitating the qualitative and quantitative detection of the instrument. The primary goal of the enzymatic hydrolysis process is to dissolve, suspend, or glue the intracellular components into the solvent in order to achieve the goal of extraction. This is achieved by using one or more enzyme solutions to dissolve the plant’s cell wall and break macromolecular chains like those of cellulose and pectin through hydrolysis. The removal of impurity interference and a rise in the concentration of the material under testing can be accomplished through purification, which facilitates qualitative and quantitative detection by the equipment. Effective pretreatment technology may increase target separation, minimize loss, and boost the extraction rate [43,44,45]. The methods for preparing and analyzing KR samples to identify isoflavone components are presented in Table 2.

### 2.1. Extraction Methods of Pueraria Isoflavone

The current research demonstrated that extraction is the basis of the research and application of KR isoflavones; research about the extraction methods mainly focuses on the extraction rate. It has always been the focus of researchers to improve the extraction efficiency of isoflavones from KR. From the perspectives of extraction rate, industrial applicability, and environmental protection, the extraction methods of KR isoflavones can be classified into traditional extraction methods and modern extraction methods. Solvent extraction is a commonly used traditional isoflavone extraction method, and the solvents include supercritical fluids and ionic liquids. Modern extraction techniques include microwave extraction, ultrasonic extraction, enzyme-assisted extraction, high-speed countercurrent chromatography, solid-phase extraction, and cloud point extraction.

#### 2.1.1. Solvent Extraction

According to the extract’s temperature, the solvent extraction process is often split into cold and hot categories. Percolation and impregnation are the two primary components of the cold extraction process. Decoction, continuous reflux, sometimes known as Soxhlet extraction, and reflux extraction are the three basic techniques used in hot extraction [79]. The most commonly used method for extracting isoflavones from KR is organic solvent extraction, followed by solvent removal. The secret to increasing extraction efficiency is by selecting the right extraction solvent. Ethanol, methanol, and water are usually used to extract isoflavones from KR. Because of its great extraction efficiency and low toxicity, ethanol is preferred. Xu [48] established a dual-spectral technique based on near-infrared (NIR) and ultraviolet-visible (UV-Vis) spectroscopy for the analyses of puerarin and total flavonoids by 30% ethanol (400 mL) to extract the puerarin and total flavonoids from KR in a water bath (70 °C) for 120 min. Huang’s [50] investigation examined the extraction of several active components using reflux extraction, ultrasound-assisted extraction, and various extraction solvents (30%, 50%, 80% methanol, and 30%, 50%, and 80% ethanol). The findings revealed no discernible difference between reflux and ultrasound, with the best extraction effect occurring when 50% methanol was utilized as the extraction solvent.

More and more green solvents are emerging to replace conventional volatile harmful solvents as science and technology advance. For instance, the extraction of Pueraria isoflavone using ILS, non-ionic surfactant micelle solution, and NADES. The application of novel green solvents in flavonoid extraction is presented in Table 3. Deep eutectic solvents (DES) are a class of green solvents commonly associated with ionic liquids because of their common properties, such as high thermal stability, low volatility, and low vapor pressure [80]. NADESs are composed of natural compounds produced by cell metabolism, and they have similar characteristics to DES. For the extraction of puerarin isoflavones, Saied A [81] created NADESs with a molar ratio of choline chloride to citric acid of 1:2. A high-performance liquid chromatography (HPLC)-diode array detector (DAD) coupled with high-resolution (HR) mass spectrometry (MS) was used to detect the extract of KR, and 10 isoflavones were detected. The amount of isoflavones in KR was measured to be 1.09 ± 0.006% overall. Puerarin was extracted from KR by Fan [82] using ultrasonic-aided extraction with IL as the extraction solvent using the response surface optimization approach. The quality of KR, the type and concentration of IL, the strength and duration of ultrasonic extraction, and all of these factors were optimized. Finally, 0.43 g of KR raw materials was added to 10 mL of 1.06 mol/L 1-butyl-3-methylimidazolium bromide aqueous solution. The best extraction effect could be reached by extracting for 27.43 min while using 480 W of ultrasonic power.

Regarding other unique isoflavones in non-KR, various research is also very extensive. Methanol, acetone, ethanol, and water were utilized by Yoshiara [87] as solvents to extract soybean isoflavones. Fifteen groups of mixed solvents were set up to extract soybean isoflavones using the simplex-centroid design method. The outcomes demonstrated that various soybean isoflavone types had various extraction efficiencies as a result of various structures. The glycoside isoflavones were effectively extracted using a combination of water, acetone, and acetonitrile. The extraction of isoflavones in the form of malonyl glucoside was positively impacted by the mixture of water, acetone, and ethanol. Jurga [88] promoted the extraction of daidzein and genistein from red clover blossoms using excipients in conjunction with ultrasound-assisted, hot reflux, and impregnation procedures. Finally, it was found that a vinylpyrrolidone-vinyl acetate copolymer facilitated the solubilization and availability of the active components of a plant extract.

#### 2.1.2. Microwave-Assisted Extraction (MAE)

The microwave-assisted extraction method, which is frequently used for trace analysis of organic compounds in solid and liquid samples [89], builds on solvent extraction. MEA utilizes the internal heating effect of the microwave (thermal stress breaks the cells, the solvent and material molecules constantly rub with the change in alternating electromagnetic fields, and the degree of cell breakage increases). It benefits from low liquid consumption, high extraction efficiency, strong selectivity, low pollution, and consistent heating, which prevents the gelatinization and agglomeration of the medicinal components. However, because of the rapid local temperature rise, it may quickly denature and inactivate the heat-sensitive material, reducing the effectiveness of the extraction. Liu [64] used response surface methodology to investigate the effects of various factors on the extraction efficiency of total flavonoids from PuerariaeLobatae Radix. The results showed that the effect of various factors on the extraction efficiency was a specific gravity liquid–solid ratio > ethanol concentration > time > power. When 42% ethanol solution was used, the liquid–solid ratio was 43:1, the microwave power was 828 W, the extraction time was 23 min, and the best extraction effect was obtained. Hu [57] optimized the process conditions, such as the volume fraction of the extraction agent, extraction time, microwave power and solid-liquid ratio after the cleaning, drying, crushing, and sieving of KR. Finally, the extraction was carried out under the conditions of a 60% ethanol volume fraction, 1:50 solid–liquid ratio, 4 min extraction time and 340 W microwave power. Shi [90] used the orthogonal test method to optimize the extraction process of Pueraria isoflavone with the extraction amount of five Pueraria isoflavones as the evaluation index. The optimum extraction process of KR was as follows: 60% ethanol reflux extraction three times, each time adding 15 times the amount of solvent, and extracting for 1 h. The results showed that the contents of 3′-hydroxy puerarin, 3′-methoxy puerarin, puerarin, daidzin, and daidzein were more than 90% of the total isoflavone content.

Cai [91] studied the microwave-assisted extraction technology of genistein and genistin from Flemingia Macrophylla with uniform design and orthogonal design. The result showed that the optimal extraction process was set as follows: microwave power of 700 W, extraction temperature of 80 °C, analytical agent ratio of 7.1 mL/g, microwave time of 120 s, liquid/solid ratio of 35 mL/g, and an extraction time of 50 min. Under the optimum conditions, the yield of genistein and genistin was 1.3804 mg/g. Li [92] used methanol reflux extraction to determine calycoisoflavones and formononetin by HPLC. The optimal microwave conditions were as follows: low-temperature heating for 5 min, thickening for 2 cm, calycoflavone 3.984 μg/g, and formononetin 42.314 μg/g.

#### 2.1.3. Ultrasound-Assisted Extraction (UAE)

The cavitation effect caused by an ultrasonic wave is used to create a cavitation gas cannon in the extraction solution in the ultrasonic-aided extraction technique, which is based on the solution extraction method. With the eruption of the shock wave, the cavitation gas cannon bursts, destroying the plant’s cell wall and increasing the solubility of the target component [93]. In contrast to other instrument-assisted extraction methods, it does not need heating, making it safe and considerate of chemicals that are sensitive to heat. Its limitation by ultrasonic attenuation factors is a drawback. For instance, the tank is too big to create an ultrasonic blank region, which negatively impacts the effectiveness of the extraction process. Punam [46] employed several extraction techniques and methanol as the extraction solvent to extract isoflavones from KR flowers. Comparisons were made between the methanol ultrasonic extract (PLs) and the methanol reflux extract (PLr). In PLs and PLr, the percentage of Pueraria isoflavone was 7.99% and 10.57%, respectively. Zhou [70] utilized 50 mL of 70% methanol as the extraction solution at 40 °C and 0.2 g of PuerariaeLobatae Radix and PuerariaeThomsonii Radix sample powder. HPLC was utilized to establish the content detection technique of 3′hydroxy puerarin, puerarin, 3′-methoxy puerarin, daidzin, puerarinapioside, and puerarin-6″-O-xyloside in KR and Pueraria thomsonii under the conditions of 250 W, 40 kHz, and ultrasonic extraction for 30 min. For the extraction of puerarin, Wu [94] utilized ethanol with a concentration of 71.35%, an extraction period of 49.08 min, and a solvent-to-material ratio of 21.72. Li [95] used single-factor and Box–Behnken experiments to improve the extraction conditions of flavonoids from “Gange 2” and KR. Two different types of Pueraria flavonoids produced yields of 3.66 mg/g and 4.08 mg/g under ideal circumstances.

#### 2.1.4. Enzyme-Assisted Extraction (EAE)

The structure of the plant’s cell wall and intercellular material will obstruct the diffusion of the target compounds into the extraction solvent during the extraction of plant isoflavones. The addition of enzymes will destroy the normal structure of cell walls and intercellular substances [96]. Common enzymes employed in the extraction of the plant’s active components include cellulase, pectinase, amylase, and protease. Single-enzyme extraction technology is being used in the actual extraction process less and less. The combination of composite enzyme methods, enzyme-assisted methods, and ultrasonic-assisted methods is being used increasingly [97]. The benefits of EAE include gentle reaction conditions, eco-friendly extraction solvents, and minimal active substance loss. However, because the enzyme is environment-sensitive, it will become inactive at high temperatures and at low pH levels, the extraction cost will increase, and the extraction efficiency will be reduced. By using the response surface approach, Liu [78] adjusted the extraction parameters for the enzyme-microwave-assisted synergistic extraction of puerarin. They looked into the three variables of cellulose dosage, microwave power, and microwave treatment time. By using the response surface method, the ideal procedure was discovered: a cellulose dosage of 190 U/g, microwave power of 450 W, and microwave treatment period of 7 s. The extraction rate for puerarin under these circumstances was 8.87 mg per 100 g. The method parameters for the enzymatic-ultrasonic extraction of puerarin from KR were improved by Huang [98]. According to the findings, the ideal conditions for extracting puerarin from KR were as follows: 0.4% cellulase addition, 70 min of enzymatic hydrolysis, a 30:1 (mL/g) liquid–solid ratio, 52% volume of ethanol, 31 min ultrasonic extraction, and 64 °C for the ultrasonic temperature. The yield of puerarin in these circumstances was 8.78 mg/g.

The extraction of soybean isoflavones has always been the focus of research. In order to increase the extraction rate of soybean isoflavones, Guo [99] selected Trichoderma viride cellulose from ten cellulases and optimized the amount of enzyme added, as well as the enzymatic hydrolysis temperature, enzymatic hydrolysis time, substrate concentration, reaction system pH, and other parameters. The findings demonstrated that more cost-effective and efficient parameters included substrate concentrations of 0.8 to 2.0 mg/mL, enzyme dosages of 7% to 11%, a reaction system pH of 5.0, an enzymatic hydrolysis temperature of 55 °C, and shaking reaction times of 5 to 6 h. The overall rate of aglycone conversion and the total rate of hydrolysis of soybean isoflavone glycosides, respectively, were as high as 93.21% and 59.25%.

#### 2.1.5. Extraction Techniques Summary

The conventional solvent extraction technique is ineffective, and using more solvents increases the possibility of environmental damage. In addition to being a popular topic in current research, the emergence of NADESs offers a new option for the extraction of Pueraria isoflavones and other plant-active components. Technology for modern instrument-assisted extraction has been continuously improved. Although Pueraria isoflavone extraction efficiency and time can be increased and decreased with the use of ultrasonic, microwave, and enzyme-assisted technologies, there are still certain drawbacks. For instance, some isoflavones that readily degrade and are not heat-resistant cannot be extracted with MAE. Ultrasonic blank areas and high noise are drawbacks of UAE. The expense and easily influenced enzyme activity by the environment are the two drawbacks of EAE. As a result, the composite extraction approach is the main area of interest and development right now. A current issue is how to use it for the commercial extraction of Pueraria isoflavones.

### 2.2. Purification Techniques for Preparing Pueraria Isoflavone

In order to further obtain high-purity isoflavones, it is often necessary to separate and purify the extracted isoflavones through a complex process.

#### 2.2.1. Column Chromatography

Column chromatography is a typical technique that utilizes the various partition coefficients of various components in the stationary phase and the mobile phase. It is often used to separate and purify flavonoids from plant-active substances. C18 reverse-phase fillers, silica gel, macroporous resin, etc., are examples of fillers that are frequently employed. Its major benefits are high efficiency, low cost, and ease of operation. Yang’s method [100] for enriching puerarin and daidzein in PuerariaeLobatae Radix utilized the HPD800 macroporous adsorption resin. The purity of puerarin increased from 8.62% to 35.26% after macroporous adsorption resin enrichment, and the purity of daidzein grew from 0.276% to 10.17%. The ability of AB-8, NKA, and X-5 macroporous adsorption resins to adsorb the total flavonoids from KR was examined by Wang [101]. The yield of the total flavonoids was 76.22%, and the results indicated that AB-8 resin was the optimum adsorbent for separation and purification. The extracting solution of KR was combined with the 0.1 M Ni^2+^ solution by Pan [102], separated using the dry silica gel column technique, and eluted using a gradient of chloroform and methanol. Finally, more than 95% of genistein and daidzein were pure. The ultimate extraction rate was 2.09% and was achieved by Yang [66] using a 70% ethanol solution microwave-assisted extraction, C18 chromatographic column separation and purification, and diode array detection. Additionally, semi-preparative HPLC is a quick and efficient separation technique for high-purity puerarin isoflavones. Using semi-preparative reverse-phase HPLC, Punam [46] isolated and purified isoflavones from the flowers of KR. When isoflavones from KR flowers were extracted using various techniques, the concentration ranged from 7.99% to 10.57%. An innovative technique for extracting and separating isoflavones from KR was created and confirmed by Li [103]. The separation of isoflavones occurred using countercurrent chromatography and semi-preparative liquid chromatography, as well as ultrasound-assisted extraction of the isoflavones based on ionic liquids. In the end, 500 g of KR yielded daidzein-4′, 7-diglucoside (42.2 mg), puerarin 6′-O-xyloside (88.3 mg), and 3′-hydroxypuerarin (48.5 mg) with purities of 98.2%, 96.3%, and 97.1%, respectively.

#### 2.2.2. High-Speed Countercurrent Chromatography (HSCCC)

HSCCC is a countercurrent chromatography-based separation method that primarily enhances the resolution, injection, and separation time of CCC. It uses chromatography with liquid–liquid separation [54,104]. It combines the benefits of liquid–liquid extraction and distribution chromatography, replaces the solid stationary phase in the conventional separation method with liquid, and can be extensively employed in the separation and purification of various components in natural mixtures. It has recently become one of the hotspots in the study of effective separation techniques [54,105]. High-speed countercurrent chromatography can be used to separate natural antioxidants because traditional solid–liquid separation cannot prevent the irreversible adsorption loss of the solid carrier matrix utilized in the chromatographic column. The monomer compounds were extracted by Liu [106] using a two-phase solvent system of ethyl acetate, ethanol, and water (4.0:0.5:3.0, *v*/*v*/*v*). The monomer compounds were then separated and purified by high-speed countercurrent chromatography and assessed by high-performance liquid chromatography. In the end, three compounds were obtained: puerarin (90.60%), genistin (99.00%), and irisflorentin (91.73%). The puerarin extract was chosen as the research object by He [107], who used the HSCCC ethyl acetate: n-butanol: water (2:1:3, *v*/*v*/*v*) solvent system at a speed of 900 r/min with a flow rate of 1.5 mL/min and a separation temperature of 25 °C under the conditions of separation and purification of high-purity puerarin samples by ultraviolet absorption spectroscopy, finally obtaining a purity of more than 98%, with an expanded uncertainty of less than 1% of the puerarin standard sample.

Calycosin and formononetin were successfully isolated from Astragalus membranaceus by Pan [108] and purified using high-speed countercurrent chromatography. Calycosin was purified using a two-phase solvent system of n-hexane, ethyl acetate, ethanol, and water (3:5:3:5), while formononetin was purified using a two-phase solvent system of n-hexane, ethyl acetate, ethanol, and water (4:5:4:5). The analysis revealed that formononetin 2.0 mg had a purity of 98.9% and calycosin 1.3 mg had a purity of 95.8%.

#### 2.2.3. Solid-Phase Extraction (SPE)

The goal of solid-phase extraction (SPE) technology is to separate and enrich the component(s) to be tested by using solid adsorbents to adsorb the component(s) to be tested in liquid samples and then eluted with eluent to achieve the separation and enrichment of the components to be tested. For the extraction and separation of Pueraria isoflavone, SPE columns such as the C18 column and HLB column are frequently utilized. To extract isoflavones from KR, Zhu [47] developed a magnetic ZIF-8 pressurized capillary electrochromatography (pCEC) method. This method was used to separate and detect the isoflavones puerarin, daidzin, and daidzein in KR. The limit of detection (LOD) of puerarin, daidzein, and daidzein was 0.02, 0.03, and 0.03 μg/mL, respectively. The average recovery was in the range of 98.5–100.3% and an RSD of < 4.0%. Using 36 mL of a 50% ethanol solution and 4 mL of hydrochloric acid, Hou [109] applied 0.5 g of TBHQ to 0.1 g of the KR samples while employing nitrogen protection and a water bath heating reflux for 120 min. The extract was extracted using an HLB solid-phase extraction column, followed by an elution step using 5 mL of a 20% methanol solution and 10 mL of methanol. Eleven flavonoid aglycones were extracted and purified at a flow rate of 1–2 mL/min.

Qiao [110] separated the material on a Diamonsil C18 (250 mm 4.6 mm, 5 m) column with methanol-0.01% KH_2_PO_4_ (28:72) after treating it with a Hyper Sep C18 solid-phase extraction column. The column temperature was 40 °C, the volume flow rate was 1.0 mL/min, and the detection wavelength was 250 nm for the mobile phase. The SPE method is the foundation of dispersive solid-phase extraction (d-SPE). After removing the interference with complete oscillation or centrifugation, the solid adsorbent is introduced directly to the solvent. Due to their ease of use and sparing use of solvents, these two techniques have garnered considerable interest.

#### 2.2.4. Cloud Point Extraction (CPE)

The new technique, cloud point extraction (CPE), also known as micellar-mediated extraction (MMe), is used to separate chemicals that are lipophilic from those that are water-soluble. It is mostly accomplished by adding surfactants to the solution that needs to be separated, followed by adjustments to the electrolyte, pH, temperature, and other variables [111]. Jiang [112] employed a Phenomenex C18 column, non-ionic surfactant Triton X-114 as an extractant, methanol-0.25% glacial acetic acid solution as the mobile phase, gradient elution, and a UV detector for detection at a wavelength of 250 nm. The content approach was used to calculate the amounts of the five isoflavones, puerarin, daidzin, genistin, daidzein, and genistein, in KR. Chi [113] investigated the ideal extraction conditions for an aqueous two-phase extraction of Pueraria isoflavone using the CRE method. The outcomes demonstrated that the optimal conditions for the separation and purification of the total flavonoids from KR were an aqueous two-phase system with an ethanol mass fraction of 50.14% and a dipotassium hydrogen phosphate mass fraction of 22.67% in a specific range of mass concentrations. The distribution coefficient of the total flavonoids in KR in the aqueous two-phase system increases with increasing mass concentration of the total flavonoids in KR. When the Pueraria crude extract mass concentration was greater than 0.014 g/mL, the variance coefficient declined. The extraction rate is mostly unaffected by the changes in mass concentration under ideal extraction circumstances. The extraction rate reached 99.23%, and the distribution coefficient of the system’s total flavonoids was 35.99. A technique for the simultaneous separation and determination of puerarin and daidzin was developed by Qu [51]. By combining CPE and the concentration with an HPLC technique, genistein, daidzein, genistin, and formononetin were identified in Pueraria. The optimal extraction conditions were as follows: a surfactant Triton X-100 concentration of 0.07 g/mL, NaCl addition of 0.6 g, a liquid–solid ratio of 80:1 (mL/g), and an equilibrium time of 40 min. Under the optimal conditions, the total maximum extraction amount of six isoflavones reached 8.92 mg/g. The CPE method can avoid the use of a large number of organic solvents and does not require the use of large and complex instruments; rather, it only needs to be completed in the plugged centrifuge tube. However, the separation principle has requirements for both the matrix and the target compound, which limits the application range of the CPE method.

#### 2.2.5. Purification Techniques Summary

Because it is inexpensive and simple to use, column chromatography can be widely employed in industrial production; however, it has drawback-low purifying effectiveness. SPE and QuEChERS have developed quickly in recent years, with one of the main development paths being the study and use of novel materials for the effective extraction of KR. HSCCC has effective separation skills and can decrease the usage of organic solvents; however, the price of the necessary equipment is too costly and makes it unsuitable for promotion. Although the CPE approach does not require the use of big or complicated apparatus and can reduce the amount of organic solvents used, its range of applications is limited. Therefore, it is important to think about ways to increase separation efficiency while lowering the extraction and separation costs. This supports the expansion of the use of Pueraria isoflavones in the food and pharmaceutical industries.

## 3. Analytical Methods

In order to study the content and biochemical properties of Pueraria isoflavone, it is necessary to identify and quantitatively analyze the extracted and purified compounds. The various techniques for detecting isoflavones in KR and its products are summarized in Table 4. The most commonly used methods for the detection of Pueraria isoflavone are thin-layer chromatography (TLC) [114], LC [115], capillary electrophoresis (CE) [116], UV, MS, etc. In order to improve the sensitivity and accuracy of the detection results, liquid chromatography is often used in combination with other technologies, including LC-UV/Vis, LC-MS, LC-MS/MS, etc.

### 3.1. LC Coupled with UV

In order to assess a substance qualitatively and quantitatively, UV/Vis uses the substance’s molecules or ions to absorb light in a certain wavelength range. For online quantitative monitoring, Xu [48] integrated near-infrared (NIR) and ultraviolet-visible (UV-Vis) spectroscopy. The liquid material from the MAE extraction was separated by Hu [57] using a high-speed centrifuge. The constituent parts of the extract were separated using thin-layer chromatography, and qualitative and quantitative analyses of the extract were carried out using UV/Vis. Preliminarily, it may be determined that the mother nucleus structure of the Pueraria flavonoids in the Dabie Mountains is primarily composed of isoflavone, dihydroflavone, and dihydroflavonols. This conclusion is based on the peak form of the UV-visible methanol spectrum of the KR extract. Although UV/Vis testing is quick and inexpensive, it can only detect substances that absorb in the ultraviolet and visible ranges, and it is easily interfered with by impurities. As a result, the combination’s influence on detection is minimal. By utilizing the separation properties of LC, the combination with LC can increase the accuracy of the detection results. The UV and LC information of Pueraria isoflavones is shown in Table 5. High-performance liquid chromatography was employed by Li [77] to quantify the puerarin content of KR. The stationary phase was the C18 column, the mobile phase was methanol–water (25:75), the flow rate was 1.0 mL/min, and the detection wavelength was 250 nm. Five batches from four different manufacturing regions had a puerarin content that ranged from 2.13% to 5.61%. Huang [50] employed a C18 column as the stationary phase, acetonitrile-0.05% formic acid as the mobile phase, with a flow rate of 0.2 mL/min, a column temperature of 30 °C, a sample volume of 2 L, gradient elution, and a detection wavelength of 250 nm. Principal component analysis and fingerprint library identification were performed. Pueraria glycoside was among the six major differential components identified. It was demonstrated that environmental influences from various production areas might influence the accumulation of secondary metabolites in KR by the discovery of soybean saponin and soyasapogenol in KR.

Li [117] established a high-performance liquid chromatography method for the simultaneous determination of soyasapogenol, genistein, formononetin, and biochanin A in red clover extract. The gradient elution was carried out using the Bridge C18 chromatographic column, with methanol (A) and 10 mmoL disodium hydrogen phosphate solution (B) serving as the mobile phases. The flow rate was 1 mL/min, and the sample volume was 10 L. A 270 nm detecting wavelength was chosen. The recovery rate was finally 96–100.10%. Lin [118] used ethanol–water (3:1, *v*/*v*) to extract from soybean, separated by the Acquity UPLC ^®^ HSS T3 column, using methanol and 0.1% formic acid solution as the mobile phase for gradient elution. The contents of genistein, daidzein, and glycitein in soybeans were determined by UPLC-MS/MS. The LODs were 0.5, 0.4, and 0.1 mg/kg, respectively. The average recovery was 79.3–107%, and the RSD was 1.3–6.8%.

### 3.2. LC Coupled with MS

When combining two separation and detection methods, its sensitivity is higher than that of liquid chromatography alone, and its selectivity and specificity are also superior. It also overcomes the lack of standards in liquid chromatography. The MS information of Pueraria isoflavones is shown in Table 6. In 45 different food types, including 22 different liquid and 23 different solid processed food types, Ahn [49] reported a range of plant-active components. The stationary phase was a Cadenza CL-C18-column (3100 mm, 3 m). The mobile phase was composed of a 95% acetonitrile solution containing 2 mM ammonium formate and 0.2% formic acid and a formic acid solution containing 0.2% ammonium formate. Additionally, a flow rate of 0.5 mL per minute was used for the gradient elution. Finally, the food had a puerarin level of 13.6–23.9 mg/g. Hou [109] used LC-MS/MS to determine the content of 11 flavonoid aglycones in the SPE extract of KR. The RP-18 column was used as the stationary phase, and the methanol–water system (daidzin, daidzein, formononetin) or methanol-0.15% formic acid solution system (the other eight flavonoid glycosides and 11 flavonoid aglycones) was used as the mobile phase for gradient elution. The recovery rate of 11 flavonoid aglycones was 75.0–94.0%, and the RSD was 3.5–12%. Shen [119] established the LC-MS/MS qualitative analysis method and HPLC analysis method for the content of puerarin in cold-clearing heat granules. The average recovery of puerarin was 100.09%. Wu [120] used Q-ExactiveLC-MS to collect data from Tongmai Jiangtang capsules (TJC). The 15 batches of TJC samples were detected, and the fingerprints were established. The average recovery of puerarin was 95.30–103.56%, and the mass fraction was 5.374–8.805 mg/g.

### 3.3. Pressurized Capillary Electrochromatography (pCEC)

Pressurized capillary electrochromatography (pCEC) is a new micro-separation technology combining high-performance liquid chromatography (HPLC) and capillary electrophoresis (CE). It not only has a high column efficiency and selectivity [121] but also has the advantages of low consumption with the mobile phase and sample, less environmental pollution, and a high performance–price ratio. It is especially suitable for the analysis of trace, multi-component, and complex-matrix components [122]. By using pressurized capillary electrochromatography, Zhang [61] developed a method for the detection of nine isoflavones in KR foods. He then used this method to investigate the isoflavone content of various KR foods. Nine isoflavones were isolated from KR food using a 20% ethanol solution that contained 0.1% formic acid and was purified using a PRiME HLB column. The voltage intensity was set at +2 kV, and the mobile phase was composed of acetonitrile-15 mmol/L potassium phosphate buffer (15:85, *v*/*v*). In the concentration range of 10.0–200.0 g/mL, the linearity of the nine isoflavone standard solutions was positive. The detection limit was 0.5–1.0 mg/kg, the limit of quantitation was 2.0–5.0 mg/kg, and the recovery rate was 86.2–98.6%. For the separation and detection of three isoflavones (puerarin, daidzin, and daidzein) in KR, Zhu [47] developed a magnetic ZIF-8-pressure capillary electrochromatography (pCEC) method. The results showed that the components recovered at a rate of between 98.5% and 100.3%, with a 4.0% RSD.

### 3.4. Analytical Methods Summary

In the current detection method, TLC has little personnel and equipment needs. It is inaccurate and requires lengthy experimental processes. The sensitivity and speed of ELISA are advantages; however, the cross-contamination reaction is a disadvantage. A novel method known as pCEC combines the benefits of capillary electrophoresis with the great selectivity of HPLC. In recent years, it has become one of chromatography’s hot sites. However, it is still in the development stage, with few applications, and there are problems, such as a small sample load and difficult cleaning. LC-UV/Vis and LC-MS are widely used in the detection and analysis of Pueraria isoflavones. The sensitivity and selectivity of MS have been improved as a result of the development of HRMS, TOF MS, ion-trap MS, and orbital trap. Improved detection techniques can also help with standardization, reliability, and food quality control, as well as the chemical analysis of natural goods.

## 4. Outlook

### 4.1. Application Value and Health Benefits of KR

The nutritional profile of KR is extensive and diverse, encompassing flavonoids, dietary fiber, starch, and various trace elements [123]. The primary constituent of KR is starch, with its levels reaching approximately 40% [124]. Of the functional components, kudzu starch is frequently utilized as a functional food for the management of type 2 diabetes mellitus (T2DM) [125]. Studies have demonstrated that there are variations in the thermodynamic properties, pasting properties, solubility, swelling, and structural characteristics of the starches extracted from different KR varieties [126]. KR is recognized for its high fiber content, which includes both soluble dietary fiber (SDF) and insoluble dietary fiber (IDF), with respective contents of 30% and 10%. In addition to dietary starch and fiber, KR is a source of isoflavones, particularly puerarin (comprising approximately 60% of the total isoflavones) and, to a lesser extent, daidzein and daidzin [127,128,129,130]. Pueraria isoflavone has been shown to confer numerous health benefits on various bodily systems, including the brain, liver, heart, kidney, bone, stomach, muscle, skin, and reproductive system [131]. Zhou et al.’s estimates indicate that puerarin and daidzein exhibit comparable antioxidant activity [132]. Liu et al.’s findings suggest that puerarin mitigates depression-like behavior in ovariectomized rats by activating the cAMP-CREB-BDNF signaling pathway [133]. Nrf2 serves as a critical regulator of cellular defense against diverse forms of oxidative damage. Yu and Wang’s research demonstrates that KR can counteract CD-induced Nrf2 inhibition, thereby preventing autophagy suppression and NLRP3 inflammasome activation [134,135]. GUO found that puerarin can inhibit inflammation and apoptosis through HDAC1/HDAC3 signaling, thereby alleviating streptozotocin (STZ)-induced osteoporosis in rats [136]. There are also rich, grand elements, such as Ca and Mg, and profitable elements, such as the trace elements Zn, Cu, and Fe, in KR [137]. These elements have many benefits for the brain, liver, heart, kidneys, stomach, etc.

### 4.2. Current Trends and Future Perspectives

Many researchers have now experimented with various techniques for extracting, purifying, and analyzing Pueraria isoflavone. The development of health foods using Pueraria as the primary raw material, as well as the use of Pueraria isoflavone and the plant’s major active element, have both benefited from these studies. In terms of extraction, some researchers combine new techniques to increase the extraction efficiency of Pueraria isoflavone. These techniques include enzyme extraction, UAE, MAE, and heating reflux. More and more green solvents, such as ILS, NADES, etc., have been used to extract Pueraria isoflavone, with promising results, in an effort to limit the usage of organic solvents. This will also become one of the primary research paths in the future. Column chromatography is commonly used for the separation and purification of Pueraria isoflavone. One of the recent research areas for more effective separation is HSCCC. It can separate and purify the different components found in natural mixtures and has the advantages of partition chromatography and liquid–liquid extraction. It has a promising future for applications. In terms of analysis and detection, UV/Vis is the most common. In order to improve the accuracy of the detection results, UV/Vis is generally combined with LC to overcome its disadvantage of being easily impacted by impurities. To improve sensitivity and the capacity for quantitative analysis, several studies also use LC-MS or LC-MS/MS to detect Pueraria isoflavone. However, the majority of these projects take place in laboratories. More research and guidance are still required to adapt it to actual factory production.

Future studies on the separation, purification, analysis, and detection of Pueraria isoflavone may focus on the following aspects:

For the extraction and purification processes, it is particularly important to develop novel, eco-friendly green solvents and auxiliary extraction instruments. The creation and production of high-purity puerarin isoflavone may play a vital role in the subsequent stage of research and health food manufacturing.

The demand for the detection of puerarin isoflavone in health foods is steadily increasing day by day with product development and manufacturing. Next, attention should be given to the development of pretreatment techniques for various health food substrates. New spectral, chromatographic, and mass spectrometry methods are also being developed to meet the demands of fast detection of Pueraria isoflavone contents in a variety of substrates. The combined use of multiple analytical instruments can help to increase the sensitivity and accuracy of detection.

## 5. Conclusions

KR’s potential as a nutrition source and its economic value cannot be undermined. Pueraria isoflavone is the main active constituent of the functional food KR and has a wide range of biological activities. The cornerstone for Pueraria isoflavones’ use in functional foods and in-depth research is the technology for extraction, purification, and detection. Research in recent years has shown that the purity and extraction efficiencies of Pueraria isoflavone are being improved by using more efficient extraction and separation methods as well as environmentally friendly extraction solvents. The precision and accuracy of the method for the determination of isoflavone contents in KR are continuously improving. The detection procedure is also developing in the direction of being rapid, simple, of high sensitivity, strong specificity, and low cost. We hope that this review can enhance people’s interest in the use of isoflavones in KR as a nutritional food.

## Figures and Tables

**Figure 1 molecules-28-06577-f001:**
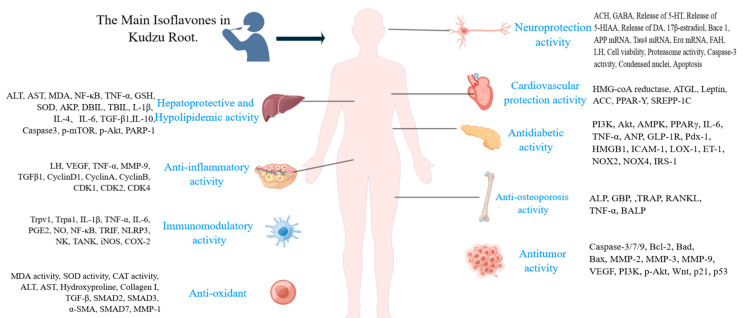
Bioactivities of main isoflavones in kudzu root and their underlying mechanisms.

**Figure 2 molecules-28-06577-f002:**
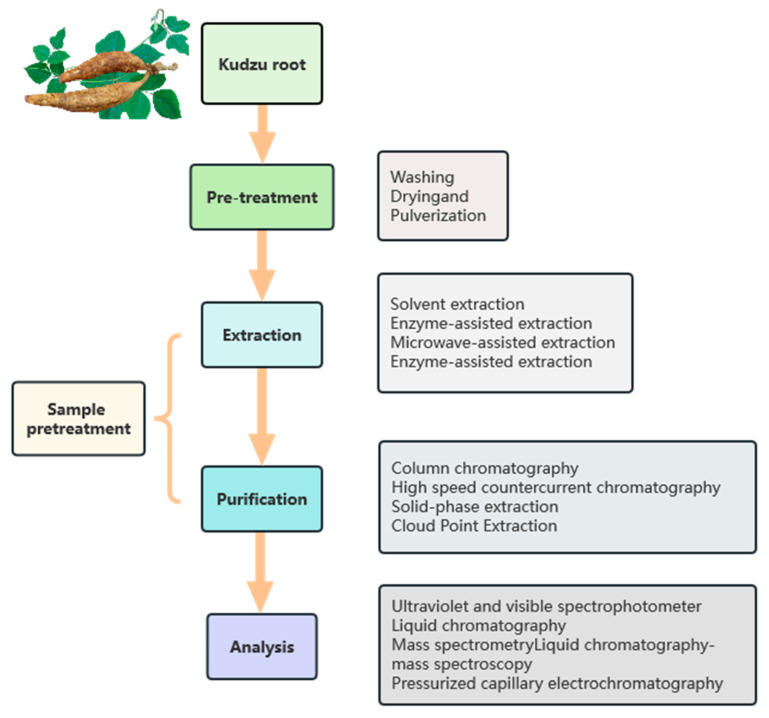
The procedure of extraction, purification, and structural analysis of isoflavones in kudzu root.

**Table 1 molecules-28-06577-t001:** The main isoflavones in kudzu root [42].

Name	Chemical Formula	Structural Formula	Content (μg/g)
Puerarin	C_21_H_20_O_9_	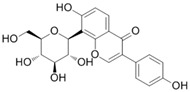	4.28–76.10
Daidzein	C_15_H_19_O_4_	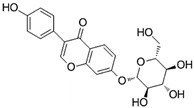	0.36–16.48
Daidzin	C_21_H_12_O_9_	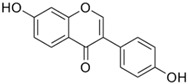	0.05–6.74
3′ hydroxyPuerarin	C_21_H_20_O_10_	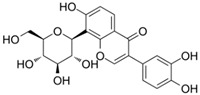	0.20–20.61
Genistein	C_15_H_10_O_5_	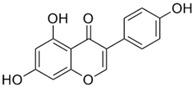	-
Genistin	C_21_H_20_O_10_	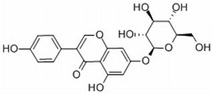	7.63–51.43
Formononetin	C_16_H_12_O_4_	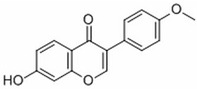	-
6″-O-Malonylgenistin	C_24_H_22_O_13_	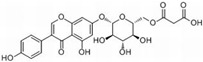	-

-: Not reported.

**Table 2 molecules-28-06577-t002:** Summary of the various techniques used to prepare analytical kudzu root for the determination of their isoflavones.

Analytes	Extraction Method	Extraction Solvent	Condition	Separation and Purification	Analysis Method	Productive Rate (%)	References
Isoflavones	Sonication extract (PLs) and reflux extract (PLr)	Methanol	Ultrasonic bath (frequency of 40 kHz; Power Sonic 520 W, at room temperature for 2 h.)	Semi-preparative reversed-phase HPLC	HPLC	7.99–10.57	[46]
Puerarin, daidzin, daidzein	Refluxing extraction	80% ethanol	Refluxed in a water bath at 80 °C for 2 h	MSPE Magnetic solid-phase extraction	ZIF-8-pressurized capillary electrochromatography (pCEC)	-	[47]
Puerarin and total flavonoids	Immersion method	30% ethanol	Water bath (70 °C)	-	UPLC-MS, NIR, and UV-Vis portable	-	[48]
Nine isoflavones	Ultrasonic-assisted extraction	Methanol	Sonicated at 40 °C for 20 min	-	HPLC	-	[8]
puerarin	SPE	80% methanol	Sonicated for 30 min	SPE	LC-MS/MS	-	[49]
Six isoflavones	Refluxing extraction and ultrasonic-assisted extraction	50% methanol	Ultrasonic bath (frequency of 40 kHz; Power Sonic 120 W)	-	UPLC	-	[50]
Isoflavones	Ultrasonic-assisted extraction based on NADES	NADESs: choline chloride and citric acid at a 1:2 molar ratio	Ultrasonic bath (frequency of 37 kHz; Power Sonic 580 W, at 60 °C for 3 h.)	Reversed stationary phase column	HPLC-DAD	1.09 ± 0.006	[51]
Isoflavone	Ultrasonic-assisted extraction	Water or ethanol 65°	Amplitude = 65% nominal power, cycle = 1, 40 °C ± 1 °C	-	HPLC-PDA	-	[52]
Puerarin and daidzein	Ultrasonic-assisted extraction	Methanol-glacial acetic acid (100:1)	Ultrasonic bath (frequency of 40 kHz; Power Sonic 100 W)	-	Non-aqueous capillary electrophoresis (NCAE)	-	[53]
Puerarin	Ultrasonic-assisted extraction	70% ethanol	Ultrasonic bath (Power Sonic 350 W at 60 °C for 3 h.)	Acid hydrolysis	HPLC-IR	-	[54]
Puerarin and daidzein	Ultrasonic-assisted extraction	Water	-	-	HPLC	-	[55]
Total flavonoids	Ultrasonic-assisted extraction	35% ethanol	Ultrasonic bath (frequency of 53 kHz; Power Sonic 200 W)	-	UV	2.76	[56]
Flavonoids	Microwave-assisted extraction	60% ethanol,	Microwave power 340 W for 4 min	-	UV	-	[57]
Six isoflavones	Refluxing extraction	30% ethanol	-	-	HPLC	-	[58]
Puerarin	Ultrasonic-assisted extraction	70% ethanol	Ultrasonic bath (frequency of 42 kHz; Power Sonic 70 W)	-	UV	-	[59]
Flavonoids	Ultrasonic-assisted extraction	75% ethanol	Ultrasonic extraction 30 min	-	UV	-	[60]
Nine isoflavones	SPE	20% ethanol solution (containing 0.1% formic acid)	-	-	pCEC	-	[61]
Puerarin and daidzein	Ultrasonic-assisted extraction	Ethanol	-	-	Differential pulse voltammetry	-	[62]
flavonoids	Refluxing extraction	60% ethanol	Reflux extraction 1.5 h	-	UV-Vis	-	[63]
Flavonoids	Microwave-assisted extraction	42% ethanol	Microwave power 828 W for 23 min	-	UV	11.74	[64]
Puerarin	Ultrasonic-assisted extraction	58% ethanol	Ultrasonic bath at 70 °C for 32 min	-	UV-Vis	-	[65]
Puerarin	Microwave-assisted extraction	70% ethanol	Mrowave 9.7 min	column C18	HPLC	-	[66]
flavonoids	Immersion method	40% ethanol	Water bath at 80 °C for 2 h	-	UV	3.06	[67]
Four isoflavones	Ultrasonic-assisted extraction	30% ethanol	Ultrasonic extraction 1 h	-	HPLC	-	[68]
Three isoflavones	Ultrasonic-assisted extraction	50% ethanol	Ultrasonic extraction 40 min	-	HPLC	-	[69]
Six isoflavones	Ultrasonic-assisted extraction	70% methanol	Ultrasonic bath (frequency of 40 kHz; Power Sonic 250 W, for 3 h.)	-	HPLC	-	[70]
Five isoflavones	Ultrasonic-assisted extraction	70% methanol	Ultrasonic extraction 1 h	-	HPLC	-	[71]
Puerarin	Ultrasonic-assisted extraction	0.6 mg/mL β-CD	Ultrasonic extraction at 40 °C for 1 h	-	Three-dimensional fluorescence spectrum	-	[72]
Flavonoids	Ultrasonic-assisted extraction	40% ethanol	Ultrasonic bath (Power Sonic 300 W for 20 min)	-	GC/MS	-	[73]
Puerarin and daidzein	Refluxing extraction	80% ethanol	-	-	HPLC	-	[74]
Puerarin	Microwave-assisted ionic liquid extraction	1.0 mol/L ionic liquids	Microwave power 400 W for 8 min	-	UV	-	[75]
Four isoflavones	Immersion method	30% ethanol	-	-	HPLC	-	[76]
Total flavonoids and puerarin	Refluxing extraction	Methanol	Heat reflux extraction 1 h	-	UV-Vis and HPLC	-	[77]
Puerarin	Microwave-assisted enzymatic extraction technology	Cellulose dose 190 U/g	Microwave power 450 W for 7 s	-	UV	8.87	[78]

-: Not reported.

**Table 3 molecules-28-06577-t003:** Application of novel green solvents in flavonoid extraction.

Title Compounds	Species	Composition	Auxiliary Extraction	Extraction Effect	References
Daidzein, genistein, puerarin	NADESs	ChCl/citric acid	-	NADESs extract had higher antioxidant activity than methanol extract and significantly reduced the degradation of isoflavones.	[83]
Puerarin	NADESs	L-Pro/malic acid	-	The extraction amount of NADESs was 2.2 times higher than that of water and also significantly higher than that of methanol. The bioavailability of the extract was 323% of the aqueous extract.	[84]
Puerarin	IL	1-normal-butyl-3- methylimidazolium chloride	MAE	The extraction rate of puerarin was 4.201%, which was three times higher than that of the traditional extraction method.	[75,85]
Flavonoids	IL	1-Butyl-3-methylimidazolium bromide	UAE	The extraction amount of pueraria flavonoids was 774.95 mg/g.	[86]
Puerarin isoflavones	NADESs	Choline chloride to citric acid of 1:2	-	The amount of isoflavones in KR was measured to be 1.09 ± 0.006% overall.	[81]
Puerarin	IL	1-butyl-3-methylimidazolium bromide aqueous solution	UAE	The proposed ILUAE offered shorter extraction time and remarkably higher efficiencies	[82]

-: Not reported.

**Table 4 molecules-28-06577-t004:** Various analytical methods for the determination of isoflavones in kudzu root.

Sample Matrix/Source	Analytes	Instrument Type	Stationary Phase	Mobile Phase	Flow Rate; Injection	Determination	LOD/LOQ	Recovery (%)	RSD (%)	References
PuerariaeFlos	Isoflavones	HPLC	C18-column-column (4.6 × 250 mm, 5 μm)	A:water B:acetonitrile	1 mL/min; 20 µL	245 nm	LOD:0.0014 mg/600 µL LOQ:0.0036 mg/600 µL	98.41	0.79	[46]
Pueraria lobata	Puerarin, daidzin, daidzein	pCEC	EP-100–20/45–3-C18 capillary column	A:45%methanol B:55% 17.5 mM sodium dihydrogen phosphate (pH 4.0)	0.08 mL/min	250 nm	LOD:0.02–0.03 µg/mL	98.5–100.3	<4.0	[47]
PuerariaeLobatae Radix	Nine isoflavones	HPLC	ZORBAX Eclipse XDB-C18 column (4.6 mm × 250 mm, 5 µm)	A:0.1% formic acid-water B:acetonitrile	1 mL/min; 10 µL			100.3–101.1	0.33–1.3	[8]
Kudzu food	Puerarin	LC-MS/MS	Cadenza CL-C18 column (3 × 100 mm, 3 µm)	A:ammonium formate in 0.2% formic acid B:2 mM ammonium formate and 0.2% formic acid in 95% acetonitrile	0.5 mL/min; 5 µL		LOD:0.013 µg/mL LOQ:0.063 µg/mL	83.3–108.2		[49]
PuerariaeLobatae Radix	Six isoflavones	UPLC	BEH C18 column (2.1 mm × 50 mm, 1.7 µm)	A:acetonitrile B:0.05% formic acid	0.2 mL/min; 2 µL	250 nm		97.78–99.63	1.0–2.3	[50]
Kudzu roots (KR)	Isoflavones	HPLC-DAD	Poroshell 120 EC-C18-column (3.0 mm × 100 mm, 2.7 µm)	A:containing 0.1% (*v*/*v*) acetic acid in the water B:containing 0.1% acetic acid (*v*/*v*)	0.7 mL/min; 5 µL	245 nm				[51]
Pueraria lobata	Puerarin and daidzein	NCAE	Uncoated fused silica capillary column 50 cm × 75 µm ID	90 mmol/L-1 sodium cholate-3.0% acetic acid-15% acetonitrile in methanol	Injection pressure: 50 mbar Injection time: 5 s	254 nm	LOD:0.4 µg/mL LOQ:0.2 µg/mL	96.72; 97.26	1.94; 2.17	[53]
Pueraria lobata	Puerarin	HPLC-IR	Pgrandsil C18 column (5 µm, 4.6 mm × 250 mm)	Methanol: 36% acetic acid:water = 25:3:72 (*v*/*v*)	1.0 mL/min; 10 µL	250 nm				[54]
Pueraria lobata	Puerarin and daidzein	HPLC	Spherigel C18 column (250 mm × 4.6 mm, 5 µm)	A:methanol B:wate	1.0 mL/min; 10 mL	250 nm		99.87; 100.32	0.70; 1.80	[55]
Pueraria lobata and Pueraria thomsonii	Six isoflavones	HPLC	C18 column (250 mm × 4.6 mm, 3.5 µm),	A:methanol B:0.2% acetic acid	0.8 mL/min, 10 µL	250 nm		98.0–103.4		[58]
Pueraria lobata	Puerarin	UV		A:phosphoric acid-water B:acetonitrile				99.56	1.14	[59]
Pueraria lobata and Pueraria thomsonii	Flavonoids	UV				250 nm		103.29	2.90	[60]
Kudzu food	Nine isoflavones	pCEC	C18 column (100 µm × 45 cm, 3 μm);	Acetonitrile-15 mmol/L buffer solution of potassium phosphate (15:85, *v*/*v*)	40 µL	230 nm	LOD:0.5~1.0 mg/kg LOQ:2.0~5.0 mg/kg	86.2~98.6	1.9~5.3	[61]
Pueraria lobata	Puerarin and daidzein	Differential pulse voltammetry				Ep = 0.600 V		75.31–87.24		[62]
Roots, stems, leaves and flowers of pueraria lobata	Flavonoids	UV-Vis				250 nm		99.96	0.89	[63]
Pueraria lobata	Puerarin	HPLC	C18 column (150 mm × 4.6 mm × 5 µm)	A:methanol B:water	1 mL/min	250 nm		97.3–99.2	3.72–6.21	[66]
Pueraria lobata and Pueraria thomsonii	Four isoflavones	HPLC	YMC-Pack ODS-A C18-column (250 mm × 4.6 mm, 5 µm)	A:water B:methanol	1.0 mL/min; 10 µL	254 nm		96.52–100.22	0.75–1.84	[68]
Pueraria lobata	Three isoflavones	HPLC	Symmetry C18 column (4.6 mm × 250 mm, 5 µm)	A:methanol B:water	0.7 mL/min	250 nm		99.0–109.1	0.99–1.93	[69]
Pueraria lobata and Pueraria thomsonii	Six isoflavones	HPLC	Poroshell 120 EC-C18 (150 mm × 4.6 mm, 4 µm)	A:0.1% phosphoric acid B:acetonitrile	1.0 mL/min; 10 µL	250 nm		97.40~101.93	1.40~2.85	[70]
Pueraria lobata	Five isoflavones	HPLC	Phenomenex C18 column (250 mm × 4.6 mm, 5 µm)	A:methanol B:water	1.0 mL/min	260 nm		96.0–100.1	0.68–1.17	[71]
Kudzu food	Puerarin	Three-dimensional fluorescence spectrum				λex/λem = 340 nm/466 nm	LOD: 1.27 × 10^−3^ µg/mL	95.82~100.33		[72]
Pueraria lobata	Flavonoids	GC/MS	HP-5 (30 m × 0.25 mm × 0.25 µm)	Helium gas	1.0 mL/min; 1 µL	510 nm		98.60	1.86	[73]
Pueraria lobata	Puerarin and daidzein	HPLC	Agilent chromatographic column (4.6 mm × 250 mm)	A:phosphoric acid-water B:acetonitrile	1 mL/min; 10 µL	203 nm		99.56–100.96	1.98–2.13	[74]
Pueraria lobata	Four isoflavones	HPLC	C18 column (250 mm × 4.6 mm, 5 µm)	A:acetonitrile B:0.1% formic acid-water	0.8 mL/min; 10 µL	254 nm		97.6~103.6	<2.0	[76]
Pueraria lobata	Total flavonoids and puerarin	UV-Vis and HPLC		Methanol-water (25:75)	1.0 mL/min	250 nm		94.16–98.4	1.42–2.38	[77]

**Table 5 molecules-28-06577-t005:** The UV and LC information of Pueraria isoflavones [115].

Name	Detecting Wavelength (nm)	Retention Time (min)
Puerarin	222.8	39.790
Daidzein	206.3	58.684
Daidzin	226.3	52.262
3′ hydroxyPuerarin	222.8	29.208
Genistein	229.8	115.292
Genistin	226.3	76.748
Formononetin	253.5	112.505

**Table 6 molecules-28-06577-t006:** The MS information of Pueraria isoflavones.

Name	Retention Time (min)	Theoretical Molecular Weight	Precise Molecular Weight	Fragment Ions
Puerarin	11.71	547.1437	547.1437	415.1007, 325.0706, 295.0600, 267.0662, 233.3677, 189.0642
Daidzein	19.83	253.0506	253.0510	224.0472, 208.0538, 196.0535, 135.0113 241.0444, 217.0496
Daidzin	9.59	415.1034	415.1023	295.0602, 227.0499, 267.0659
3′ hydroxyPuerarin	9.18	431.0983	431.0968	331.0552, 269.0448
Genistein	20.71_5_	269.0455	269.0457	241.0497, 213.0521, 199.0390, 197.0624, 185.0614, 141.0770, 181.0660, 169.0658
Genistin	15.42	431.0983	431.0971	311.0564, 269.0439, 241.0495

## Data Availability

Not applicable.

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
