# Peer review of "Advances in Extraction, Purification, and Analysis Techniques of the Main Components of Kudzu Root: A Comprehensive Review"

_molecules, 2023, doi:10.3390/molecules28186577_

Round 1

Reviewer 1 Report

I have some questioned regarding the articles 

1.      Page 1 line 43, it should be space between “naturalmaterial”

2.      The tittle of the manuscript is extraction, purification and analysis techniques of the main components of Kudzu root, but author focusing on isoflavone only without explained it as main components. No quantitative result of how much isoflavone content in Kudzu Root

3.      Figure 1 is not clear, author only write pharmacological activities with receptor that responsible for the activity or what?

4.      Table 2, better to add column about percentage of analytes obtain (recoveries) using extraction method that has been done

5.      Author should conclude which extraction method is better in obtaining isoflavone from Kudzu root rather than just reveal result of each extraction method before moving to next subchapter (purification techniques). The same suggestion apply to purification techniques and analytical method section

Author Response

In accordance with Reviewer 1’s suggestion:

Q1. Page 1 line 43, it should be space between “naturalmaterial”

A1: Thanks for your suggestion. The spelling mistake has been corrected now. The corrected sentence is as follows: Most importantly, as an excellent natural material for the development of new health foods, KR is extensively used in the food and health products industry and has great market potential.

Q2. The tittle of the manuscript is extraction, purification and analysis techniques of the main components of Kudzu root, but author focusing on isoflavone only without explained it as main components. No quantitative result of how much isoflavone content in Kudzu Root

A2: Thank you for the comment on this issue. We supplemented the relevant discussion on Pueraria isoflavones as the main active components in Kudzu root, and supplemented the specific content of various isoflavones in Table 1. The paragraph in manuscript appears as follows:

The common ingredients of KR include flavonoids, starch, cellulose fiber, protein, and etc. Among them, the best effect ingredients are isoflavones. Isoflavones are part of a large family of secondary plant metabolites called flavonoids, they are dietary phytoestrogens occurring naturally in legumes. A reliable source of isoflavones is Pueraria lobata, according to pertinent study studies. The mass fraction of puerarin can range from 1.58% to 7.68%, whereas the overall mass fraction of isoflavones in Pueraria lobata can range from 6.2% to 17.0%[7,8]. Currently, it has been revealed that more than 50 different types of puerarin isoflavone have unambiguous structures, Puerarin[9], Daidzein, Daidzin, Puerarinxyloside[10], 3'-Hydroxy Puerarin, Genistein, Genistin, Biochanin A, Formononetin [11], 6”-O-Malonylgenistin [12], 6″-O - α - d - Glucopyranosylpuerarin[13],Calycosin[14],4-Hydroxy-7-Hydroxymethyl-6-Methoxyisoflavone[15], 4′,6-Dimethoxy-8-Hydroxy-7-Hydroxymethyl Isoflavone [16], 3'-MethoxypuerarinA, 3'-MethoxypuerarinB[17], etc. The main isoflavones in KR was displayed in Table 1.

 Table 1. The main isoflavones in Kudzu Root.[42]

Name

Chemical formula

Structural formula

Content(μg/g)

Puerarin

C21H20O9

4.28~76.10

Daidzein

C15H19O4

0.36~16.48

Daidzin

C21H12O9

0.05~6.74

3’ hydroxyPuerarin

C21H20O10

0.20~20.61

Genistein

C15H10O5

-

Genistin

C21H20O10

7.63~51.43

Formononetin

C16H12O4

-

6”-O-Malonylgenistin

C24H22O13

-

-: Not reported.

Q3. Figure 1 is not clear, author only write pharmacological activities with receptor that responsible for the activity or what?

A3: Thanks for your suggestion. We have replaced the images with improved quality. Figure 1 mainly describes the biological activity and related mechanism of Pueraria isoflavones.

Figure 1. Bioactivities of main isoflavones in KR and their underlying mechanisms.

Q4. Table 2, better to add column about percentage of analytes obtain (recoveries) using extraction method that has been done

A4: Thank you very much for this suggestion. We have modified it according to your suggestion. Increase the “Productive rate” column in the original table.

Analytes

Extraction method

Extraction solvent

Condition

Separation and purification

Analysis method

Productive rate

(%)

References

isoflavones

sonication extract (PLs) and reflux extract (PLr)

methanol

ultrasonic bath (frequency of 40 kHz; Power Sonic 520W,at room temperature for 2 h.)

semi-preparative reversed-phase HPLC

HPLC

7.99 -10.57

[46]

puerarin, daidzin, daidzein

refluxing extraction

80% ethanol

refluxed in a water bath at 80℃ for 2 h

MSPE

magnetic solid-phase extraction

ZIF-8-pressurized capillary electrochromatography (pCEC)

-

[47]

puerarin and total flavonoids

immersion method

30% ethanol

water bath (70 ℃)

-

UPLC-MS ,NIR and UV–Vis portable

-

[48]

9 isoflavones

ultrasonic-assisted extraction

methanol

sonicated at 40 ℃ for 20 min

-

HPLC

-

[49]

puerarin

SPE

80% methanol

sonicated for 30 min

SPE

LC-MS/MS

-

[50]

6 isoflavones

refluxing extraction and ultrasonic-assisted extraction

50% methanol

ultrasonic bath (frequency of 40kHz; Power Sonic 120W)

-

UPLC

-

[51]

isoflavones

Ultrasonic-assisted extraction based on NADES

NADESs: choline chloride and citric acid at a 1:2 molar ratio

ultrasonic bath (frequency of 37 kHz; Power Sonic 580W,at 60℃ for 3 h.)

reversed stationary phase column

HPLC-DAD

1.09 ± 0.006

[52]

isoflavone

ultrasonic-assisted extraction

water or ethanol 65°

amplitude = 65% nominal power, cycle = 1, 40 °C ±1 °C

-

HPLC-PDA

-

[53]

puerarin and daidzein

ultrasonic-assisted extraction

methanol-glacial acetic acid ( 100 ∶ 1 )

ultrasonic bath (frequency of 40kHz; Power Sonic 100W)

-

non-aqueous capillary electrophoresis(NCAE)

-

[54]

puerarin

ultrasonic-assisted extraction

70% ethanol

ultrasonic bath (Power Sonic 350W at 60℃ for 3 h.)

acid hydrolysis

HPLC-IR

-

[55]

puerarin and daidzein

ultrasonic-assisted extraction

water

-

-

HPLC

-

[56]

total flavonoids

ultrasonic-assisted extraction

35% ethanol

ultrasonic bath (frequency of 53 kHz; Power Sonic 200 W)

-

UV

2.76

[57]

flavonoids

microwave-assisted extraction

60% ethanol,

microwave power 340 W for 4 min

-

UV

-

[58]

6 isoflavones

refluxing extraction

30% ethanol

-

-

HPLC

-

[59]

puerarin

ultrasonic-assisted extraction

70% ethanol

ultrasonic bath (frequency of 42 kHz; Power Sonic 70 W)

-

UV

-

[60]

flavonoids

ultrasonic-assisted extraction

75 % ethanol

ultrasonic extraction 30 min

-

UV

-

[61]

9 isoflavones

SPE

20 % ethanol solution ( containing 0.1 % formic acid )

-

-

pCEC

-

[62]

puerarin and daidzein

ultrasonic-assisted extraction

ethanol

-

-

differential pulse voltammetry

-

[63]

flavonoids

refluxing extraction

60% ethanol

reflux extraction 1.5 h

-

UV-Vis

-

[64]

flavonoids

microwave-assisted extraction

42% ethanol

microwave power 828 W for 23 min

-

UV

11.74

[65]

puerarin

ultrasonic-assisted extraction

58% ethanol

ultrasonic bath at 70℃ for 32 min

-

UV-Vis

-

[66]

puerarin

microwave-assisted extraction

70% ethanol

microwave 9.7 min

column C18

HPLC

-

[67]

flavonoids

immersion method

40% ethanol

water bath at 80 °C for 2 h

-

UV

3.06

[68]

4 isoflavones

ultrasonic-assisted extraction

30%ethanol

ultrasonic extraction 1 h

-

HPLC

-

[69]

3 isoflavones

ultrasonic-assisted extraction

50% ethanol

ultrasonic extraction 40 min

-

HPLC

-

[70]

6 isoflavones

ultrasonic-assisted extraction

70% methanol

ultrasonic bath (frequency of 40 kHz; Power Sonic 250W, for 3 h.)

-

HPLC

-

[71]

5 isoflavones

ultrasonic-assisted extraction

70% methanol

ultrasonic extraction 1h

-

HPLC

-

[72]

puerarin

ultrasonic-assisted extraction

0.6 mg/mLβ-CD

ultrasonic extraction at 40°C for 1h

-

three-dimensional fluorescence spectrum

-

[73]

flavonoids

ultrasonic-assisted extraction

40% ethanol

ultrasonic bath (Power Sonic 300 W for 20 min)

-

GC/MS

-

[74]

puerarin and daidzein

refluxing extraction

80% ethanol

-

-

HPLC

-

[75]

puerarin

microwave-assisted ionic liquid extraction

1.0 mol/L ionic liquids

microwave power 400 W for 8 min

-

UV

-

[76]

4 isoflavones

immersion method

30% ethanol

-

-

HPLC

-

[77]

total flavonoidsand

refluxing extraction

methanol

heat reflux extraction 1 h

-

UV-Vis and HPLC

-

[78]

puerarin

microwave-assisted enzymatic extraction technology

cellulose dose 190U/g

microwave power 450 W for 7s

-

UV

8.87

[79]

Q5. Author should conclude which extraction method is better in obtaining isoflavone from Kudzu root rather than just reveal result of each extraction method before moving to next subchapter (purification techniques). The same suggestion apply to purification techniques and analytical method section

A5: Thanks for your helpful comments. At the end of each summary of extraction, purification and analysis, we summarized each part.

2.1.5. Extraction summary

The conventional solvent extraction technique is ineffective, and using more sol-vents increases the possibility of environmental damage. In addition to being a popular topic in current research, the emergence of NADESs offers a new option for the extraction of Pueraria isoflavones and other plant active components. Technology for mod-ern instrument-assisted extraction has been continuously improved. Although Pueraria isoflavone extraction efficiency and time can be increased and decreased with the use of ultrasonic, microwave, and enzyme-assisted technologies, there are still certain drawbacks. For instance, some isoflavones that readily degrade and are not heat-resistant cannot be extracted with MAE. Ultrasonic blank area and high noise are drawbacks of UAE. The expense and easily influenced enzyme activity by the environment are the two drawbacks of EAE. As a result, the composite extraction approach is the main area of interest and development right now. A current issue is how to use it for the commercial extraction of Pueraria isoflavones.

2.2.5 Purification techniques summary

Because it is inexpensive and simple to use, column chromatography can be widely employed in industrial production, but it has a drawback—low purifying effective-ness. SPE and QuEChERS have developed quickly in recent years, with one of the main development paths being the study and use of novel materials for the effective extraction of KR. HSCCC has effective separation skills. And can decrease the usage of organic solvents, but the price of the necessary equipment is too costly and makes it un-suitable for promotion. Although the CPE approach does not require the use of big or complicated apparatus and can reduce the amount of organic solvents used, its range of applications is limited. Therefore, it is important to think about ways to increase separation efficiency while lowering extraction and separation costs. This supports the expansion of the use of Pueraria isoflavones in the food and pharmaceutical industries.

3.4. Analytical methods summary

In the current detection method, TLC has little personnel and equipment needs, it is inaccurate and requires lengthy experimental processes. The sensitivity and speed of ELISA are advantages, but the cross-contamination reaction is a disadvantage. A novel method known as pCEC combines the benefits of capillary electrophoresis with the great selectivity of HPLC. In recent years, it has become one of the chromatography's hot sites. However, it is still in the development stage, with few applications, and there are problems such as small sample load and difficult cleaning. LC-UV / Vis and LC-MS are widely used in the detection and analysis of Pueraria isoflavones. The sensitivity and selectivity of MS have been improved as a result of the development of HRMS, TOF MS, Ion Trap MS, and orbital trap. Improved detection techniques can also help with standardization, reliability, and food quality control, as well as chemical analysis of natural goods

Reviewer 2 Report

The manuscript submitted for evaluation is a review of advances in kudzu root phytochemical research. This applies primarily to methodological issues related to typical analytical procedures such as extraction and identification of bioactive compounds. This is a valuable study summarizing the achievements in this field over the last several years, when a clear progress in research on this valuable plant was observed. Observable through the development of analytical techniques related primarily to the development of chromatography, in particular detection techniques and the development of unconventional processing methods.

The present manuscript in this respect has been properly compiled with a thorough review of the available literature which includes 131 cited items. Nevertheless, I believe that some points could be added to the content of the current manuscript for the benefit of its upgrading.

First of all, kudzu is a plant that contains many valuable bioactive substances, the consumption of which has a positive effect on human health and has a preventive effect in many chronic diseases. The authors could include an extended description of many valuable biological activities, which in the current form of the manuscript have been described quite casually. I suggest adding a dedicated paragraph describing examples of biological activity. In particular, the recently discovered aid in addiction treatment.

I also believe that paragraph 2. Sample pretreatment methods can be extended. The authors mention the possible use of pre-treatment of the plant matrix for better extraction of analytes. It is possible to specify here, for example, procedures related to enzymatic treatment or hydrolysis. In paragraph 2.1.1 I would like more details on the use of deep eutectic fluids. It is quite popular today and fast growing sector of green chemistry. Including an additional table with NADES examples would be appreciated. Paragraphs 3.1 and 3.2 should be supplemented with a list of spectral properties such as UV-Vis maxima and m/z and MS/MS characteristic fragments.

I believe that after a positive response to my comments, the whole manuscript will be significantly improved and will be a valuable source material to help other scientists dealing with this subject.

Author Response

In accordance with Reviewer 2’s suggestion:

Q1: First of all, kudzu is a plant that contains many valuable bioactive substances, the consumption of which has a positive effect on human health and has a preventive effect in many chronic diseases. The authors could include an extended description of many valuable biological activities, which in the current form of the manuscript have been described quite casually. I suggest adding a dedicated paragraph describing examples of biological activity. In particular, the recently discovered aid in addiction treatment.

A1: Thanks for your suggestion. We added a dedicated paragraph describing examples of biological activity. The paragraph in manuscript appears as follows:

Research has revealed that puerarin isoflavone has positive effects on common health issues like diabetes, hypertension, and hyperlipidemia. It has been established that puerarin, genistein, daidzein and daidzin are efficient treatments for hypertension[33–38]. Pueraria extract, according to Li et al., can lower blood pressure in mice fed a high-salt diet through modulating gut flora[37]. According to Yang et al., pueraria is effective in treating physical illness-es that co-occur with depression[38]. It covers post-stroke depression as well as de-pression caused by coexisting conditions such diabetes, coronary heart disease, migraines, Parkinson's disease, and others. KR have potential in an alcohol intake reduction[39], and thus in the long-term prevention of alcohol addiction[40]. Zhou et al. gave alcoholic male Wistar rats puerarin, daidzein, and puerarin[41]. The findings demonstrated that daidzin and puerarin were able to dramatically raise the levels of two types of ghrelin, which in turn decreased alcohol consumption.

Q2: I also believe that paragraph 2. Sample pretreatment methods can be extended. The authors mention the possible use of pre-treatment of the plant matrix for better extraction of analytes. It is possible to specify here, for example, procedures related to enzymatic treatment or hydrolysis.

A2: Thanks for your helpful comments. We expanded the paragraph 2. And briefly described the procedures related to enzyme treatment and hydrolysis in the extraction process. The paragraph in manuscript appears as follows:

Sample pretreatment refers to the separation of components from complex systems by enzymatic hydrolysis, extraction, enrichment, purification and concentration. It is a process of removing impurity interference, increasing the concentration of the component to be tested, and facilitating the qualitative and quantitative detection of the instrument. The primary goal of the enzymatic hydrolysis process is to dissolve, suspend, or glue the intracellular components into the solvent in order to achieve the goal of extraction. This is done by using one or more enzyme solutions to dissolve the plant cell wall and break macromolecular chains like cellulose and pectin through hydrolysis. The removal of impurity interference and a rise in the concentration of the material under test can be accomplished through purification, which facilitates qualitative and quantitative detection by the equipment. Effective pre-treatment technology may increase target separation, minimize loss, and boost extraction rate[43–45].The methods for preparing and analyzing KR samples to identify iso-flavone components are presented in Table2.

Q3: In paragraph 2.1.1 I would like more details on the use of deep eutectic fluids. It is quite popular today and fast growing sector of green chemistry. Including an additional table with NADES examples would be appreciated.

A3: Thank you for your precious suggestions. We supplemented the application of new green solvents in the extraction process of Pueraria isoflavones and the related information of DES. The paragraph in manuscript appears as follows:

More and more green solvents are emerging to replace conventional volatile harmful solvents as science and technology advance. The application of novel green solvents in flavonoids extraction are presented in Table3. For instance, the extraction of Pueraria isoflavone using ILS, nonionic surfactant micelle solution, and NADES. Deep eutectic solvents (DES) is a class of green solvents commonly associated with ionic liquids because of common properties, such as high thermal stability, low volatility, and low vapor pres-sure[81]. NADES are composed of natural com-pounds produced by cell metabolism, and they have similar characteristics to DES. For the extraction of puerarin isoflavones, Saied A[82] created NADESs with a molar ratio of choline chloride to citric acid of 1: 2. High performance liquid chromatography (HPLC)-diode array detector (DAD) coupled with high resolution (HR) mass spectrometry (MS) was used to detect the extract of KR, and 10 isoflavones were detected. The amount of isoflavones in KR was measured to be 1.09 ± 0.006% overall. Puerarin was extracted from KR by Fan[83] using ultrasonic aided extraction with IL as the extraction solvent using the response surface optimization approach. The quality of KR, the kind and concentration of IL, the strength and duration of ultrasonic extraction, and all of these factors were optimized. Finally, 0.43 g of KR raw materials were added to 10 mL of 1.06 mol/L 1-butyl-3-methylimidazolium bromide aqueous solution. The best extraction effect could be reached by extracting for 27.43 min while using 480 W of ultrasonic power.

Table 3. Application of novel green solvents in flavonoids extraction.

Title

compounds

Species

Composition

Auxiliary

extraction

Extraction  effect

References

Daidzein, genistein, puerarin

NADESs

ChCl / citric acid

-

NADESs extract had higher antioxidant activity than methanol extract and significantly reduced the degradation of isoflavones.

[84]

Puerarin

NADESs

L-Pro / malic acid

-

The extraction amount of NADESs was 2.2 times higher than that of water, and also significantly higher than that of methanol. The bioavailability of the extract was 323 % of the aqueous extract.

[85]

Puerarin

IL

1-     normal-butyl-3-

methylimidazolium chloride

MAE

The extraction rate of puerarin was 4.201 %, which was three times higher than that of the traditional extraction method.

[76,86]

Flavonoids

IL

1-Butyl-3-methylimidazolium bromide

UAE

The extraction amount of pueraria flavonoids was 774.95 mg / g.

[87]

Puerarin

isoflavones

NADESs

choline chloride to citric acid of 1: 2

-

The amount of isoflavones in KR was measured to be 1.09 ± 0.006% overall.

[82]

Puerarin

IL

1-butyl-3-methylimidazolium bromide aqueous solution

UAE

The  proposed ILUAE offered shorter extraction time and remarkable higher efficiencies

[83]

-: Not reported.

Q4: Paragraphs 3.1 and 3.2 should be supplemented with a list of spectral properties such as UV-Vis maxima and m/z and MS/MS characteristic fragments.

A4: Thanks for the suggestion of improvement. We supplemented the relevant information of the target compounds in Table 5 and Table 6.

Table5. The UV and LC information of Pueraria isoflavones.[118]

Name

Detecting wavelength

(nm)

Retention time

(min)

Puerarin

222.8

39.790

Daidzein

206.3

58.684

Daidzin

226.3

52.262

3’ hydroxyPuerarin

222.8

29.208

Genistein

229.8

115.292

Genistin

226.3

76.748

Formononetin

253.5

112.505

Table6. The MS information of Pueraria isoflavones.

Name

Retention time

(min)

Theoretical molecular weight

Precise molecular weight

Fragment ions

Puerarin

11.71

547.1437

547.1437

415.1007, 325.0706,  295.0600, 267.0662,  233.3677, 189.0642

Daidzein

19.83

253.0506

253.0510

224.0472, 208.0538, 196.0535, 135.0113

241.0444, 217.0496

Daidzin

9.59

415.1034

415.1023

295.0602, 227.0499,

267.0659

3’ hydroxyPuerarin

9.18

431.0983

431.0968

331.0552, 269.0448

Genistein

20.715

269.0455

269.0457

241.0497, 213.0521, 199.0390, 197.0624, 185.0614, 141.0770,

181.0660, 169.0658

Genistin

15.42

431.0983

431.0971

311.0564, 269.0439,

241.0495

Round 2

Reviewer 1 Report

author has revised all my major concern

Reviewer 2 Report

The authors responded positively to my comments, which, in my opinion, significantly improved the manuscript. I Recommend accepting the manuscript in its current form.